# Epidemiology and Drug Susceptibility of Nontuberculous Mycobacteria in the Province of Pavia (Northern Italy): An Overview

**DOI:** 10.3390/microorganisms13112547

**Published:** 2025-11-07

**Authors:** Mariangela Siciliano, Francesco Amisano, Jessica Bagnarino, Giulia Grassia, Patrizia Cambieri, Fausto Baldanti, Vincenzina Monzillo, Daniela Barbarini

**Affiliations:** 1Department of Microbiology and Virology, Fondazione IRCCS Policlinico San Matteo, 27100 Pavia, Italy; m.siciliano@smatteo.pv.it (M.S.); j.bagnarino@smatteo.pv.it (J.B.); g.grassia@smatteo.pv.it (G.G.); p.cambieri@smatteo.pv.it (P.C.); f.baldanti@smatteo.pv.it (F.B.); vincenzinamonzillo@gmail.com (V.M.); d.barbarini@smatteo.pv.it (D.B.); 2Department of Hematology 3—Cell Factory and Center for Advanced Cellular Therapies, Fondazione IRCCS Policlinico San Matteo, 27100 Pavia, Italy; 3Specialization School of Microbiology and Virology, University of Pavia, 27100 Pavia, Italy; 4Department of Clinical-Surgical, Diagnostic, and Pediatric Sciences, University of Pavia, 27100 Pavia, Italy

**Keywords:** antimicrobial susceptibility, public health surveillance, *M. avium* complex, *M. abscessus*, pulmonary disease, extrapulmonary infections, environment

## Abstract

Nontuberculous mycobacteria (NTM) represent a heterogeneous group of environmental opportunistic pathogens that have emerged particularly in immunocompromised individuals and patients with underlying pulmonary disorders. NTM infections primarily affect the lungs, but can also manifest as lymphadenitis, skin and soft tissue infections, and disseminated disease. This retrospective study took into consideration 425 NTM-positive samples collected between May 2011 and December 2023, analyzed by sample type, sex, and age group (0–17, 18–49, 50–65, >65 years). Antimicrobial susceptibility analysis was performed on the 223 NTM strains with greater pathogenic power and most frequently isolated, from 2016 to 2023. Pulmonary NTM disease (NTM-PD) infections were most prevalent in patients over 65 years (52.1%), while extrapulmonary NTM disease (NTM-EPD) occurred most frequently in the 0–17 age group (56.4%). Women were slightly more affected (54.4%) than men (45.6%), with the highest incidence in female individuals over 65 years old. The most frequently isolated NTM species was the *Mycobacterium avium* complex (MAC) (47% of isolates). Antimicrobial susceptibility testing of 223 isolates from 2016 to 2023 revealed species-specific resistance patterns, with high susceptibility to clarithromycin in MAC (94.7%) and *Mycobacterium chelonae* (100%), but notable resistance in *Mycobacterium abscessus* complex (MABC). The increasing incidence of NTM infections underscores the need for improved diagnostic techniques and targeted treatment strategies.

## 1. Introduction

Nontuberculous mycobacteria (NTM) are a group of bacteria belonging to the genus *Mycobacterium*, which includes species other than *Mycobacterium tuberculosis* complex (MTC), the causative agent of tuberculosis [1]. Currently, NTM comprise over 200 globally ubiquitous species in both natural and anthropogenic environments. NTM are generally acquired from the environment through ingestion, inhalation, and contact [2]; person-to-person transmission of NTM is considered rare [3].

NTM pulmonary disease (NTM-PD) is the most common clinical manifestation and has become a major global public health concern due to the sharp increase in incidence and prevalence worldwide. Most pulmonary infections occur in patients with predisposing factors such as chronic obstructive pulmonary disease (COPD), bronchiectasis, cystic fibrosis, or a history of TB. These conditions make individuals particularly susceptible to colonization [4]. Lung infections are primarily caused by *Mycobacterium avium* complex (MAC) and *Mycobacterium abscessus* complex (MABC), followed by *Mycobacterium xenopi*, *Mycobacterium fortuitum*, and *Mycobacterium kansasii* [5]. MAC includes slow-growing mycobacteria (SGM) such as *Mycobacterium avium*, *Mycobacterium intracellulare*, and *Mycobacterium chimaera*, the species most frequently associated with clinical disease. MABC includes rapidly growing mycobacteria (RGM), divided into three subspecies: *M. abscessus* subsp. *abscessus*, *M. abscessus* subsp. *bolletii*, and *M. abscessus* subsp. *massiliense*. NTM also cause extrapulmonary diseases (NTM-EPD) such as cervical lymphadenitis in children, skin, soft tissue and prosthesis infections [6,7,8], bone and joint complications, and disseminated disease, especially in individuals with compromised immune systems [9].

Epidemiological studies demonstrate a heterogeneous distribution of NTM-PD. Population-based data from high-income countries suggest incidence rates ranging from approximately 4 to 20 cases per 100,000 persons per year, depending on the methodology and definitions applied [10]. In the United States, a large claims-based analysis estimated a mean annual incidence of 20.1 per 100,000 population between 2010 and 2019, with a significant upward trend over time [11]. The prevalence of NTM-PD in Europe is lower than in North America, Japan, and Korea, but is still increasing [12,13]. Specifically, it ranged from 0.9 to 7.0 per 100,000 in the UK, from 1.3 to 13.6 in France, from 3.3 to 8.4 in Spain, from 3.9 to 8.2 in Germany, from 2.3 to 5.9 in the Netherlands, and from 3.8 to 10.4 in Italy [14,15,16].

NTM constitute a substantial clinical challenge across both diagnostic and therapeutic domains. This is largely attributable to the complexity of early detection, the high prevalence of antimicrobial resistance, and the adverse effects associated with prolonged multidrug regimens. Nevertheless, a considerable proportion of prevalent cases are presumed to remain underreported, owing to persistent diagnostic limitations and the absence of mandatory surveillance or notification systems. Given the limited epidemiological information currently available in Italy, this study was designed to provide an updated overview of the epidemiological distribution of NTM during 2011–2023 in the province of Pavia. Given the heterogeneous spectrum of NTM species, antimicrobial resistance testing was performed only on a limited number of isolates, specifically those most frequently recovered and regarded as clinically significant pathogens. This focused approach was adopted to optimize statistical reliability and to generate clinically meaningful resistance data.

## 2. Materials and Methods

### 2.1. Study Setting

This retrospective study was conducted at the Microbiology Laboratory of the Fondazione IRCCS Policlinico San Matteo, Pavia, Italy, and included all clinical samples that tested positive for mycobacteria between May 2011 and December 2023. Clinical specimens were obtained from patients with suspected mycobacterial infections as part of routine diagnostic procedures. A total of 425 NTM strains, comprising both SGM and RGM species, were considered. For comparison, 456 MTC strains isolated during the same period were also analyzed. For each patient, the first isolation of each year was included in the analysis. For each isolate, anonymized data were retrieved, including sample identification code, patient age, and sex. Temporal trends in isolation were assessed by calculating the annual distribution and isolation rates of individual NTM species, and these were compared with MTC isolates. For NTM, the distribution of positive samples was analyzed by specimen type, sex, and age group (0–17, 18–49, 50–65, and >65 years). Due to the large diversity of species and the limited number of isolates for less common NTM, antimicrobial susceptibility testing (AST) was performed only on the 223 most frequently isolated and clinically significant species (MAC, *M. kansasii*, MABC, *M. chelonae*). Minimum inhibitory concentrations (MICs) were determined using the broth microdilution method according to the Clinical and Laboratory Standards Institute (CLSI) guidelines [17]. For each drug and species, MIC_50_ and MIC_90_ values and percentages for each category (susceptibility, intermediate, and resistance) were reported.

### 2.2. Culture and Identification of NTM

Respiratory samples, including sputum, endotracheal aspirates, bronchoalveolar lavages, and lung and pleural biopsies, were examined for the diagnosis of NTM-PD. Other materials, such as urine, biopsies, cerebrospinal fluid, cavitary fluids, and pus, were processed for the diagnosis of NTM-EPD. Samples were processed following international guidelines [18]. After Kinyoun staining, decontamination using 0.25% N-acetyl-L-cysteine and 1% NaOH (NALC-NaOH) was performed, according to the MycoTB^TM^ (Copan, Brescia, Italy) manufacturer’s instructions. To improve sensitivity, samples were cultured on both solid media Löwenstein–Jensen (Termo Fisher Scientific^TM^, Waltham, MA, USA) and liquid media BD BACTEC^TM^ MGIT^TM^ 960 (Becton Dickinson, Franklin Lakes, NJ, USA). Löwenstein–Jensen cultures were incubated at 37 °C in a 5% CO_2_ atmosphere for 60 days, while MGIT^TM^ tubes were incubated in the automated BACTEC MGIT^TM^ 960 system at 37 °C for 56 days. Positive samples were identified using the commercial PCR reverse hybridization method GenoType CM/AS from 2014 onwards, and NTM-DR was implemented from 2019 onwards (Hain Lifescience/Arnika, Nehren, Germany).

### 2.3. Drug Susceptibility Testing

Broth microdilution is the recommended method by CLSI Standards [17]. For SGM, broth microdilution assays were performed using Sensititre^TM^ SLOMYCO2 Susceptibility Testing Plate assay (Thermo Fisher Scientific^TM^) on isolates previously grown on Middlebrook 7H11 agar (Liofilchem^®^, Roseto degli Abruzzi, Italy). The SLOMYCO panel included the following drugs: amikacin, clarithromycin, ciprofloxacin, doxycycline, ethambutol, ethionamide, isoniazid, linezolid, moxifloxacin, rifabutin, rifampin, streptomycin, and trimethoprim/sulfamethoxazole. For each drug, MIC values were read after 7–14 days of incubation at 35 °C. For RGM, drug susceptibility tests were conducted by using the Sensititre^TM^ Myco RAPMYCOI AST Plate assay (Thermo Fisher Scientific^TM^). The following antibiotics were tested: trimethoprim/sulfamethoxazole, ciprofloxacin, moxifloxacin, cefoxitin, cefepime, ceftriaxone, amikacin, doxycycline, tigecycline, clarithromycin, linezolid, minocycline, amoxicillin/clavulanic acid, imipenem, and tobramycin. Sensititre panel plates were incubated at 30 °C for 3–5 days. For MABC, the clarithromycin incubation period was extended to 14 days to evaluate inducible resistance to macrolides. MICs were interpreted according to the breakpoints in the CLSI document; in particular, MAC MICs were interpreted according to CLSI. “Antimycobacterial Agents and Breakpoint for Testing MAC” [17] (breakpoint criteria are available in Section A.1. Table A1); *M. xenopi* MICs were interpreted according to CLSI. “Antimycobacterial Agents and Breakpoint for Testing Slowly Growing Nontuberculous Mycobacteria Other than MAC and *M. kansasii*” [17] (breakpoint criteria are available in Section A.1. Table A2), while MABC and *M. chelonae* MICs were interpreted according to CLSI. “Antimycobacterial Agents and Breakpoint for Testing Rapidly Growing Mycobacteria” [17] (breakpoint criteria are available in Section A.1. Table A3). Notably, MACs have different amikacin breakpoints, depending on the route of administration, whether it is parenteral or inhaled.

### 2.4. Statistical Analysis

Temporal trends in the annual number of isolates were assessed using linear regression, reporting regression coefficients (β), coefficients of determination (R^2^), and corresponding *p*-values. Differences in the distribution of isolates between groups (e.g., NTM vs. MTC, pulmonary vs. extrapulmonary sources) were evaluated by comparing proportions, using the chi-square test. A two-tailed *p*-value < 0.05 was considered statistically significant. All statistical analyses were performed using MedCalc statistical software Version 22.017.

## 3. Results

The number of MTC and NTM isolates from pulmonary and extrapulmonary samples exhibited distinct temporal patterns across the four groups (Figure 1). A total of 425 NTM isolates were identified, of which 370/425 (87.1%) were recovered from respiratory specimens, while 55/425 (12.9%) originated from non-respiratory sources. In comparison, during the same period, 456 isolates of MTC were detected, with 354/456 (77.6%) derived from respiratory samples and 102/456 (22.4%) from extrapulmonary sites. NTM-PD isolates showed the largest fluctuations, ranging from 10 to 44 cases per year. A general increasing trend was observed over the study period (linear regression: β ≈ +1.8 isolates/year, R^2^ = 0.42, *p* < 0.05). Peaks occurred in 2013–2014, 2016–2017, and 2021–2022, whereas troughs were recorded in 2015, 2018, and 2020. MTC-PD isolates ranged from 13 to 45 per year, without a significant linear trend over time (β ≈ +0.3 isolates/year, R^2^ = 0.08, *p* = n.s.). NTM-EPD and MTC-EPD isolates remained consistently lower, generally ≤10 cases/year. No significant temporal variation was detected in these categories (β ≈ 0, R^2^ < 0.05, *p* = n.s.). Analysis of these data indicates a relative increase in the incidence of NTM infections compared to MTC in recent years, particularly with respect to pulmonary manifestations (*p* value < 0.05).

The distribution across age groups differed markedly between the MTC-PD and NTM-PD groups (Figure 2). In the youngest age group (0–17 years), 6 cases were classified as MTC-PD compared with 5 in the NTM-PD group. Among adults aged 18–49 years, the majority belonged to the MTC-PD group (*n* = 211), whereas the NTM-PD group had only 50 cases. Conversely, in the 50–65 and >65 age groups, NTM-PD were more prevalent (50–65 years: MTC-PD *n* = 63, NTM-PD *n* = 122; >65 years: MTC-PD *n* = 74, NTM-PD *n* = 193). A chi-square test of independence revealed a significant association between age group and disease classification (χ^2^ = 170.99, df = 3, *p* < 0.001), indicating that MTC-PD was more frequent in younger adults, while NTM-PD predominated in older age groups.

The distribution of cases across age groups differed markedly between the NTM-PD and NTM-EPD groups (Table 1). In the youngest age group (0–17 years), 1.4% were classified as NTM-PD compared with 56.4% in the NTM-EPD group. Among adults aged 18–49 years, 50 (13.5%) cases belonged to NTM-PD and 8 (14.5%) to NTM-EPD. In the 50–65 and >65 age groups, NTM-PD was predominant (50–65 years: NTM-PD *n* = 122 (33%), NTM-EPD *n* = 7 (12.7%); >65 years: NTM-PD *n* = 193 (52.1%), NTM-EPD *n* = 9 (16.4%)). A chi-square test of independence revealed a significant association between age group and disease classification (χ^2^ = 190.49, df = 3, *p* < 0.001), indicating that NTM-EPD was more frequent in the youngest age group, while NTM-PD predominated in older age groups. Specifically, NTM-PD primarily affected individuals over 65 years (52.1%), whereas NTM-EPD was more frequently observed in the 0–17 age group (56.4%).

The distribution of cases across age groups by sex is presented in Table 2. A preliminary comparison of proportions indicates that in the younger age groups (0–17 and 18–49 years), the proportions of males and females were roughly similar, whereas in the older age groups (50–65 and >65 years), females slightly outnumbered males. These patterns suggest a trend of increasing female predominance with age, although the difference appears modest in the 50–65 group and more pronounced in the >65 group. In the youngest age group (0–17 years), there were 19 males (9.8%) and 17 females (7.4%). In the 18–49 age group, 31 cases were male (16%) and 27 were female (11.7%). Among the 50–65 years group, 59 males (30.4%) and 70 females (30.3%) were recorded. In the oldest age group (>65 years), 85 males (43.8%) and 117 females (50.6%) were observed. A chi-square test of independence indicated no significant association between age group and sex distribution (χ^2^ = 3.20, df = 3, *p* = 0.36), suggesting that the proportion of males and females did not differ significantly across age groups.

Among the 425 isolates considered in the study (Figure 3), the most frequently detected species was *M. avium* (*n* = 120, 28.2%), followed by *M. intracellulare* (*n* = 67, 15.8%), while 42 (9.9%) isolates were only reported as belonging to MAC, due to the unavailability of the GenoType NTM-DR at the time of their isolation. They were followed by *M. gordonae* (*n* = 54, 12.7%) and the *M. abscessus* complex that comprised 36 isolates (8.5%). Other species, including *M. xenopi* (*n* = 27, 6.3%), *M. fortuitum* complex (*n* = 16, 3.8%), *M. chimaera* (*n* = 13, 3.0%), and *M. chelonae* (*n* = 11, 2.6%), were less frequently detected. Rare species included *M. kansasii* (*n* = 8, 1.9%), *Mycobacterium malmoense* (*n* = 7, 1.6%), and *Mycobacterium marinum* (*n* = 5, 1.2%). The remaining isolates were categorized as “Other” (*n* = 19, 4.5%). A chi-square goodness-of-fit test indicated a highly significant deviation from a uniform distribution (χ^2^ = 389.84, df = 12, *p* < 0.001), showing that certain species, particularly *M. avium*, *M. intracellulare*, and *M. gordonae*, were significantly more prevalent than others.

### Antibiotic Resistance Patterns

The isolated NTM strains exhibited species-specific antibiotic resistance patterns. AST was performed on NTM strains isolated between 2016 and 2023, and here, we report only the results of 223 strains, belonging to the most frequently represented species. SGM included in the analysis were MAC and *M. xenopi*, while the RGM considered were MABC and *M. chelonae*. Following CLSI guidelines, only antibiotics with established breakpoints were considered in this study. AST was performed on 170 MAC and 19 *M. xenopi* isolates (Table 3). For MAC, the MIC_50_ and MIC_90_ values for clarithromycin were 4 µg/mL and 8 µg/mL, respectively, with 94.7% of isolates classified as susceptible, 2.9% as intermediate, and 2.4% as resistant. Intravenous amikacin showed an MIC_50_ of 16 µg/mL and MIC_90_ of 64 µg/mL, with 60.6% susceptibility; in contrast, liposomal amikacin susceptibility was observed in 97.6% of strains. Moxifloxacin had limited activity against MAC, with 5.8% susceptible, 72.4% intermediate, and 21.8% resistant isolates. Linezolid showed minimal activity, with only 2.9% susceptibility and 87.7% resistance. For *M. xenopi*, clarithromycin exhibited strong activity with an MIC_50_ of 0.12 µg/mL and MIC_90_ of 0.25 µg/mL; 100% of isolates were susceptible. Amikacin showed an MIC_50_ of 8 µg/mL and MIC_90_ of 64 µg/mL, with 84.2% susceptibility. Moxifloxacin demonstrated good activity, with 94.7% of isolates susceptible. Linezolid was highly effective (100% susceptibility). Ciprofloxacin showed moderate activity, with 63.2% susceptibility and 36.8% intermediate. Doxycycline, trimethoprim/sulfamethoxazole, and rifampicin exhibited high resistance rates (84.2%, 84.2%, and 63.2%, respectively), whereas rifabutin remained fully active (100% susceptibility).

Analysis of MABC isolates, as shown in Table 4, revealed a high susceptibility rate to intravenous amikacin (92.8%), with an MIC_50_ of 4 µg/mL and MIC_90_ of 16 µg/mL. Linezolid susceptibility was 67.9%, with an MIC_50_ of 8 µg/mL and MIC_90_ of 32 µg/mL. Clarithromycin susceptibility was 39.3%, with an MIC_50_ of 8 µg/mL and MIC_90_ of 16 µg/mL. Susceptibility to moxifloxacin, ciprofloxacin, and doxycycline was lower (4%). No susceptibility was observed for trimethoprim/sulfamethoxazole, cefoxitin, imipenem, and tobramycin. For *M. chelonae*, a broader spectrum of susceptibility was observed. All the tested strains (100%) were susceptible to clarithromycin, intravenous amikacin, and linezolid. Susceptibility to moxifloxacin was 66.7%, while 50% of strains were susceptible to ciprofloxacin and tobramycin. Susceptibility to other drugs varied, as detailed in Table 4. Comparative analysis revealed significantly higher MIC values for MABC compared with *M. chelonae* across several antibiotics (*p* < 0.05), consistent with the observed resistance patterns (*p* < 0.01).

## 4. Discussion

Currently, in Italy, there is limited information available on the epidemiology and drug susceptibility of NTM infections. The monitoring of demographic and microbiological data related to NTM is coordinated by the Istituto Superiore di Sanità (ISS) (Italian National Institute of Health, Rome, Italy), in collaboration with a network of 42 hospital laboratories across 16 out of 20 regions, including our institution [19]. In this context, the present study delineates an extensive examination of the long-term epidemiological patterns and the antimicrobial susceptibility profiles of NTM within a specified region of Northern Italy. This analysis spans a period of 13 years and emphasizes various demographic factors, including temporal trends, sex-based differences, and age-specific variances. NTM positivity exhibited significant interannual variability, with certain periods characterized by increased detection rates and others by a decline in cases. While multiple factors may have contributed to these year-to-year fluctuations, the notable rise in both NTM-PD and NTM-EPD cases—concurrent with a decrease in MTC-PD and MTC-EPD cases during the COVID-19 pandemic—could correlate with the implementation of public health measures such as mask usage, social distancing, and school closures, which likely contributed to reduced transmission dynamics [20]. Starting in 2022, a reversal in the previously observed trend was noted, likely attributable to the progressive relaxation or removal of public health measures implemented during the COVID-19 pandemic. Regarding age-based differences, this study observed a significant disparity in positivity rates between MTC-PD and NTM-PD. MTC-PD tends to have a higher incidence and more severe clinical manifestations in children compared to adults due to several factors [21]. Tuberculosis in children is often paucibacillary, making microbiological confirmation difficult [22]. NTM-PD increases among the elderly population, particularly those aged ≥65 years. Numerous studies have documented that pulmonary NTM infections more frequently affect elderly patients [23,24]. Aging is associated with a gradual decline in both innate and adaptive immune responses [25]. Elderly individuals often present with underlying pulmonary conditions such as COPD, bronchiectasis, or previous TB-related scarring, which create a favorable environment for NTM colonization and infection. In Italy, although national surveillance is limited, several regional and multicenter studies provide valuable epidemiological insights, particularly regarding the population aged ≥ 65 years. A multicenter retrospective analysis conducted by the network IRENE (Italian Registry of Nontuberculous Mycobacteria) across 42 hospitals found that most NTM cases were observed in individuals over the age of 60, with a predominance in females (57%) and in patients with chronic pulmonary comorbidities, particularly bronchiectasis and COPD [26]. A recent study conducted by Giannoni et al. analyzed laboratory-based data from the ISS monitoring over a five-year period. The results indicated a progressive increase in NTM isolation, with age-stratified data revealing a clear overrepresentation of patients aged ≥ 60 years, suggesting increased vulnerability in older age groups [19]. Several national and multicenter studies reported the highest incidence and mortality in individuals over 65 years [27,28,29]. While in adults, pulmonary forms predominate, the highest positivity rates of NTM-EPD are most prevalent among children and adolescents. A global meta-analysis on pediatric NTM infections shows that in children (especially ages 1–5), about 71% of cases involve lymphadenitis. This indicates different infection routes—oral in children compared with inhalation in adults—and reflects the immature immune system of young children [30]. In contrast, adults and the elderly generally showed lower positivity rates. Regarding sex-based differences, a slightly higher positivity rate was observed in postmenopausal women, especially as regards NTM-PD, especially as regards non-cavitary, nodular–bronchiectatic form. After menopause, estrogen decline may impair the host’s ability to control mycobacterial infections [4,31,32]. The most common NTM that causes pulmonary disease in the province of Pavia is MAC, which is consistent with its predominance in Italy and in other parts of the world [13,33,34,35] followed by *M. gordonae*, *M. xenopi* and *M. kansasii* [36]. Among RGM, MABC is the most prevalent, followed by *M. fortuitum* and *M. chelonae* three species found to be the most representative in other epidemiological studies [37]. Water, soil, and dust are known environments where MAC can live. In homes, MAC is commonly found in tap water, bathrooms, potting and garden soil, and these can be sources of infection. MAC likely spreads from natural sources into households through water distribution systems. The global spread of pulmonary MAC disease might be influenced by human activities, since people can carry MAC on themselves and their belongings, contributing to its transmission through travel and trade. Although living environments are now cleaner and more comfortable than in the past—and medical advances have increased life expectancy—these changes may unintentionally support MAC survival by reducing microbial competition through disinfection [38].

Antibiotic resistance in NTM is an increasingly important public health concern. The therapeutic approach to NTM infections is based on combined antibiotic regimens, owing to the natural drug resistance of some NTM and the potential emergence of resistance during treatment [39]. Data obtained by broth microdilution assays showed that clarithromycin was the most effective drug for MAC, *M. xenopi*, and *M. chelonae*, while it appears to be less effective for MABC. One of the most significant problems in treating MABC infections is resistance to a wide range of antibiotics, including some first-line drugs such as macrolides. These drugs are commonly used to treat various NTM infections, but in the case of MABC, inducible resistance renders them largely ineffective [40]. As regards aminoglycosides, amikacin is the most effective drug against MABC, *M. xenopi*, and *M. chelonae*, while it is less effective against MAC. According to the treatment guidelines established by the American Thoracic Society and the Infectious Diseases Society of America, patients with advanced-stage or previously treated MAC pulmonary disease are recommended to receive intravenous aminoglycosides (streptomycin or amikacin). However, the administration of an intravascular aminoglycoside is restricted by the risk of ototoxicity and renal toxicity [41]. For these reasons, the FDA approved liposome amikacin for inhalation to deliver high concentrations of the drug to the lungs in patients with MAC infection who have not achieved negative sputum culture [42]. The addition of liposomal amikacin to standard regimens has been associated with decreased hospitalization rates. The observed effects in treating NTM-PD, especially those sustained by MAC, are attributed to its targeted drug delivery at high concentration and to the unique pharmacological properties [43]. However, the FDA has not yet approved liposomal amikacin for MABC disease. Nevertheless, in some expert centers, it is used as an off-label therapy aimed at improving outcomes for these patients, who have a condition that is inherently difficult to treat [44,45].

Moxifloxacin demonstrates variable efficacy against different mycobacterial species [46]. MAC isolates showed low susceptibility, with most strains being intermediate or resistant, whereas *M. xenopi* and *M. chelonae* were largely susceptible. MABC exhibited very high resistance. These data suggest moxifloxacin is potentially useful against *M. xenopi* and *M. chelonae* but not for MAC or MABC infections. These findings are corroborated by pharmacokinetic–pharmacodynamic studies, which suggest that moxifloxacin may be useful against *M. xenopi* and *M. chelonae* infections but not for MAC or MABC infections [47]. Linezolid resistance was pronounced in MAC, with high resistance rates reported in clinical isolates (up to 80–90%) [46], whereas *M. xenopi* and *M. chelonae* isolates were fully susceptible. MABC showed moderate susceptibility. Linezolid is primarily reserved for refractory or resistant NTM infections, particularly *M. abscessus*. Its prolonged use is restricted due to hematologic and neurologic toxicity, often emerging after 2–4 months of therapy. Therapeutic drug monitoring and close clinical follow-up are recommended to minimize adverse effects while maintaining efficacy [48]. ciprofloxacin activity was generally poor in MAC [49] and MABC [50], while *M. xenopi* and *M. chelonae* showed intermediate susceptibility [51]. For MAC and *M. chelonae*, ciprofloxacin is usually not recommended as a principal agent. For MABC, ciprofloxacin may show intermediate/variable activity but is rarely sufficient alone—consider it only within combination regimens guided by MIC results. Ciprofloxacin showed moderate in vitro activity against *M. xenopi*, with susceptibility rates higher than those of MAC but lower than for moxifloxacin. Despite some clinical use in combination regimens, current guidelines recommend moxifloxacin over ciprofloxacin due to superior potency and clinical efficacy [52]. The role of doxycycline in NTM infections is minimal. It lacks consistent in vitro activity against both SGM and RGM and is not recommended as part of empiric or guideline-based treatment regimens. Its use should be restricted to confirmed susceptible isolates, primarily of *M. chelonae*, and in specific, localized infections [18]. Similarly, trimethoprim/sulfamethoxazole does not appear to be a viable option for mycobacterial infections; data were limited for MAC and *M. xenopi*, with high resistance observed in MABC and *M. chelonae*. Rifampicin and rifabutin exhibit moderate activity against MAC and *M. xenopi*, while most RGM, including MABC and *M. chelonae*, are resistant. Rifabutin is generally preferred over rifampicin in MAC due to higher potency and better intracellular activity [53].

There are several limitations in this study. First, the NTM-PD definition was based only on microbiological criteria adapted from the American Thoracic Society and the Infectious Diseases Society of America. The lack of diagnostic test results, such as chest X-ray and chest computed tomography, is a fundamental limitation for pulmonary disease confirmation. Second, the inability to accurately classify MAC species before the use of GenoType NTM-DR represents another limitation of this study. Finally, the study did not include all strains for AST because species less frequently isolated over a 13-year period may not be representative for the purposes of the survey (low sample size). The standardized MIC microdilution method was introduced in our laboratory in 2016, and we therefore could not include results obtained previously using a different method. Antibiograms were not performed for all strains because some were not considered pathogenic or clinically relevant, as suggested by the guidelines [18]. Furthermore, before 2016, it was not customary to stock strains.

## 5. Conclusions

This study provides a comprehensive overview of NTM epidemiology and antimicrobial susceptibility in a Northern Italian province over a 13-year period. The data confirm a rising trend in NTM infections, and the distribution of NTM species aligns with global patterns, with MAC as the predominant cause of pulmonary disease. AST revealed species-specific patterns that have important clinical implications. The variability in drug susceptibility highlights the necessity of individualized treatment regimens and the potential role of emerging therapeutic strategies, such as liposomal aminoglycosides, for improving outcomes in difficult-to-treat cases. Overall, these findings emphasize the importance of accurate species identification and susceptibility-guided therapy in NTM infections. The observed epidemiological trends underscore the need for ongoing surveillance, particularly in aging populations and high-risk groups.

## Figures and Tables

**Figure 1 microorganisms-13-02547-f001:**
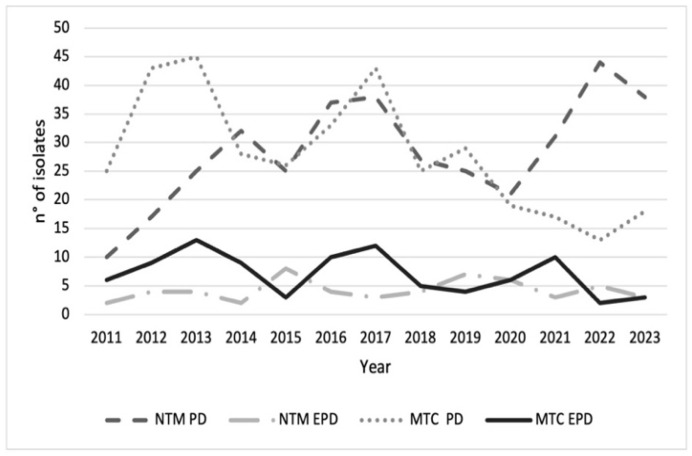
Trends of *Mycobacterium tuberculosis* complex (MTC) (*n* = 456) and nontuberculous mycobacteria (NTM) (*n* = 425) isolates for pulmonary disease (PD) and extrapulmonary disease (EPD) during 2011–2023. For each case, the analysis included only the first isolation recorded for the year. The figure indicates a relative increase in the incidence of NTM-PD compared to MTC-PD, particularly since 2020.

**Figure 2 microorganisms-13-02547-f002:**
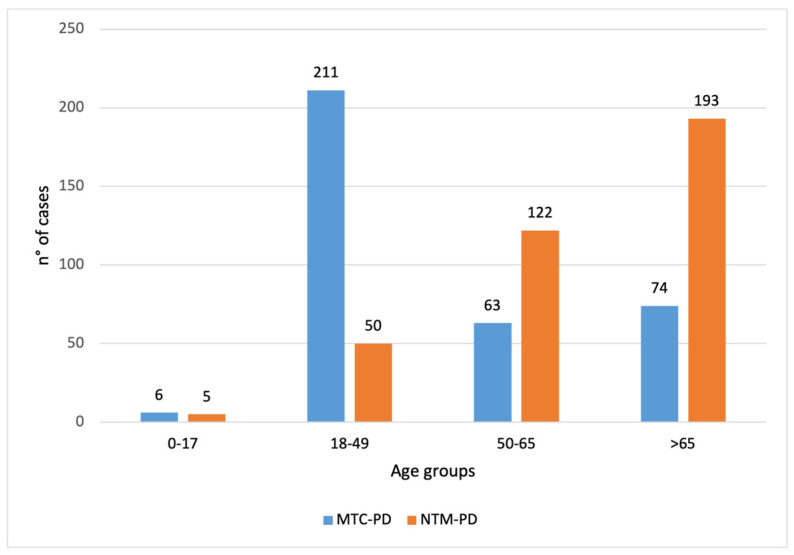
Distribution of *Mycobacterium tuberculosis* complex pulmonary disease (MTC-PD, blue) (*n* = 364/456) and nontuberculous mycobacteria pulmonary disease (NTM-PD, orange) (*n* = 370/425) positive samples by age groups (0–17, 18–49, 50–65, and >65 years). For each case, the analysis included only the first isolation recorded for the year. The figure indicates that MTC-PD was more frequent in the 18–49 age group (211 vs. 50), while NTM-PD predominated in the >65 age group (193 vs. 74).

**Figure 3 microorganisms-13-02547-f003:**
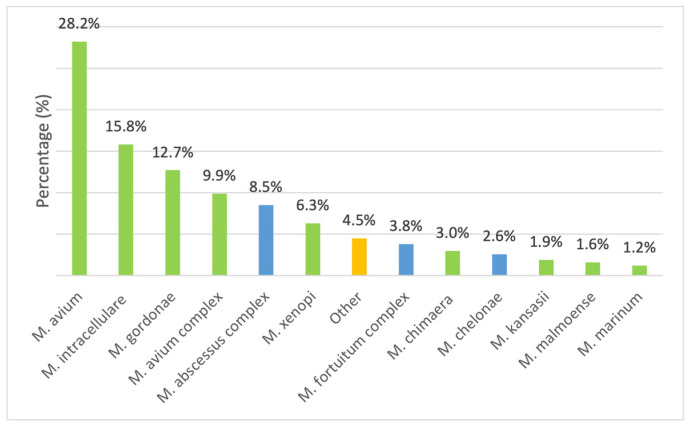
Species distribution among slow-growing mycobacteria (SGM) (green) and rapidly growing mycobacteria (RGM) (blue). The orange column labeled “other” indicates the other, less representative isolated species not included in the study. For each case, the analysis included only the first isolation recorded for the year.

**Table 1 microorganisms-13-02547-t001:** Age distribution (0–17, 18–49, 50–65, and >65 years) among nontuberculous mycobacteria pulmonary disease (NTM-PD) (*n* = 370/425) and nontuberculous mycobacteria extrapulmonary disease (NTM-EPD) (*n* = 55/425). For each case, the analysis included only the first isolation recorded for the year. NTM-PD primarily affected individuals over 65 years (52.1%), whereas NTM-EPD was more frequently observed in the 0–17 age group (56.4%).

Age Groups	NTM-PD		NTM-EPD	
	*n*.	%	*n*.	%
0–17	5	1.4	31	56.4
18–49	50	13.5	8	14.5
50–65	122	33	7	12.7
>65	193	52.1	9	16.4
**Total**	370		55	

**Table 2 microorganisms-13-02547-t002:** Gender and age distribution (0–17, 18–49, 50–65, and >65 years) among nontuberculous mycobacteria (NTM) cases. For each case, the analysis included only the first isolation recorded for the year. A chi-square test of independence indicated no significant association between age group and sex distribution (χ^2^ = 3.20, df = 3, *p* = 0.36).

Age Groups	Total		Male		Female	
	*n*.	%	*n*.	%	*n*.	%
0–17	36	8.5	19	9.8	17	7.4
18–49	58	13.6	31	16	27	11.7
50–65	129	30.4	59	30.4	70	30.3
>65	202	47.5	85	43.8	117	50.6

**Table 3 microorganisms-13-02547-t003:** Drug resistance profiles of *M. avium complex* (MAC) (*n* = 170/223) and *M. xenopi* (*n* = 19/223). Breakpoint criteria are available in Section A.1. Table A1 for MAC and Section A.2. Table A2 for *M. xenopi*. A hyphen (-) indicates the absence of a breakpoint. Amikacin interpretation refers to the intravenous resistance breakpoint. Clarithromycin (CLA), amikacin (AMI), moxifloxacin (MXF), linezolid (LZD), ciprofloxacin (CIP), doxycycline (DOX), trimethoprim/sulfamethoxazole (SXT), rifampin (RIF), rifabutin (RFB). For each case, the analysis included only the first isolation recorded for the year.

SGM (189/223)	PARAMETERS	CLA	AMI	MXF	LZD	CIP	DOX	SXT	RIF	RFB
MAC (*n* = 170/223)	MIC_50_ (µg/mL)	4	16	4	32	-	-	-	-	-
	MIC_90_ (µg/mL)	8	64	8	64	-	-	-	-	-
	Susceptible (%)	94.7	60.6	5.8	2.9	-	-	-	-	-
	Intermediate (%)	2.9	28.2	72.4	9.4	-	-	-	-	-
	Resistant (%)	2.4	11.2	21.8	87.7	-	-	-	-	-
*M. xenopi* (*n* = 19/223)	MIC_50_ (µg/mL)	0.12	8	0.5	4	1	16	8/152	2	0.25
	MIC_90_ (µg/mL)	0.25	64	1	8	2	16	8/152	8	1
	Susceptible (%)	100	84.2	94.7	100	63.2	5.3	15.8	36.8	100
	Intermediate (%)	0	5.3	5.3	0	36.8	10.5	0	0	0
	Resistant (%)	0	10.5	0	0	0	84.2	84.2	63.2	0

**Table 4 microorganisms-13-02547-t004:** Drug resistance profiles of *M. abscessus* complex (MABC) (*n* = 28/223) and *M. chelonae* (*n* = 6/223). Breakpoint criteria are available in Section A.3. Table A3. Amikacin interpretation refers to the intravenous resistance breakpoint. Clarithromycin (CLA), amikacin (AMI), moxifloxacin (MXF), linezolid (LZD), ciprofloxacin (CIP), doxycycline (DOX), trimethoprim/sulfamethoxazole (SXT), cefoxitin (FOX), imipenem (IMI), and tobramycin (TOB). For each case, the analysis included only the first isolation recorded for the year.

RGM (34/223)	PARAMETERS	CLA	AMI	MXF	LZD	CIP	DOX	SXT	FOX	IMI	TOB
MABC (*n* = 28/223)	MIC_50_ (µg/mL)	8	4	8	8	4	16	8/152	128	64	8
	MIC_90_ (µg/mL)	16	16	8	32	4	16	8/152	128	64	16
	Susceptible (%)	39.3	92.8	3.6	67.9	3.6	3.6	0	0	0	0
	Intermediate (%)	0	3.6	7.1	10.7	0	3.6	0	32.1	25	21.4
	Resistant (%)	60.7	3.6	89.3	21.4	96.4	92.8	100	67.9	75	78.6
*M. chelonae* (*n* = 6/223)	MIC_50_ (µg/mL)	0.25	2	1	2	1	16	4/76	64	16	1
	MIC_90_ (µg/mL)	0.5	16	2	8	4	16	8/152	128	64	8
	Susceptible (%)	100	100	66.7	100	50	16.7	16.7	33.3	16.7	50
	Intermediate (%)	0	0	33.3	0	33.3	16.7	0	16.7	33.3	33.3
	Resistant (%)	0	0	0	0	16.7	66.6	83.3	50	50	16.7

## Data Availability

The original contributions presented in this study are included in the article. Further inquiries can be directed to the corresponding author.

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
