# Peer review of "Epidemiology and Drug Susceptibility of Nontuberculous Mycobacteria in the Province of Pavia (Northern Italy): An Overview"

_microorganisms, 2025, doi:10.3390/microorganisms13112547_

Round 1
Reviewer 1 Report (Previous Reviewer 1)
Comments and Suggestions for Authors
This is a solid retrospective overview spanning 2011–2023 with species-level epidemiology and a focused 2016–2023 AST analysis of clinically relevant NTM. The methods follow CLSI microdilution, and the drug-specific MIC/MIC₉₀/percent susceptible data for MAC, M. xenopi, and MABC are explicitly reported (e.g., clarithromycin, amikacin including liposomal formulation moxifloxacin, linezolid, etc.). The epidemiology section properly anchors local trends against European and global estimates (rates by country and increasing prevalence), which helps contextualize the Pavia data. The Abstract clearly communicates the design and main findings. There are a few refinements I recommend before acceptance: a). Tighten the “Conclusions” to match the journal template (it currently reads like a mini-discussion and even retains a template note asking for a rewrite); b). Expand the limitations already acknowledged (species coverage, selection of isolates for AST, potential lack of clinical case adjudication for NTM-PD); c). Add minimal, high-value clarity to figure/table legends (sample denominators, breakpoints reference, and whether susceptibility calls for amikacin refer to IV or inhaled breakpoints this is noted in Table 3 caption but should be consistent across the text).
Criterion-by-criterion evaluation
1). Does the introduction provide sufficient information and include all relevant references? Perhaps. While it frames the global/European epidemiology of NTM with concrete incidence ranges and country-level comparisons, it motivates Pavia's analysis. However, it is somewhat lengthy and contains many references that the authors believe are capable of "summarizing" for more didactic purposes.
2). Is the research design appropriate? Yes. A single-center retrospective design for surveillance and AST trends at the species level is appropriate. The decision to limit AST to frequent/clinically significant species is reasonable and explicitly stated. Minor request: state any inclusion rule for "first isolate per patient per year" more prominently in the Methods (it is there; consider reiterating it in figure legends where annual counts are shown).
3). Are the methods described adequately? Yes. The study setting, time window, isolate selection, and CLSI-based microdilution are reported with citation to guidelines. Please include the exact version of the CLSI document used for NTM (you cite the CLSI “Susceptibility Testing of Mycobacteria…” in the references) and specify which breakpoint tables were applied by species/drug, corresponding to what is already implied for amikacin.
4). Are the results clearly presented? Yes (with a minor improvement). Drug-specific MICâ‚…â‚€/MIC₉₀ and %S/I/R for MAC and M. xenopi are clearly tabulated, and MABC data are summarized. Ensure that all tables have the denominator (n) in the header and that any hyphens (“–”) for missing breakpoints are consistently noted.
5). Are the conclusions supported by the results? They could be improved. The content of the conclusions is well aligned (species distribution consistent with global standards; species-specific AST implications). However, the section currently includes a template prompt from the editor requesting a rewrite. Please convert it into a concise and objective response to the study objective (a short paragraph) and move any prospective comments to the Discussion section.
6). Are all figures and tables clear and well-presented? Yes (minor polishing). The content is readable. Add: (a) explicit notation of the source of the drug-based breakpoint below each table; (b) confirm that "amikacin susceptibility" refers to the intravenous vs. inhaled liposomal breakpoint, when applicable. Your text clarifies this; please reflect this clarity in the table footnotes.
Detailed Strengths
a). Contextualization: European incidence comparisons and trend framing are accurate and current, which helps generalize local findings.
b). Methodological Transparency: The retrospective cohort period, isolate selection rules, and CLSI microdilution are stated.
c). Clinically actionable AST granularity: Reporting MICâ‚…â‚€/MIC₉₀ with %S/I/R by species—especially clarithromycin and amikacin for MAC and M. xenopi, and noting the limited utility of fluoroquinolones/linezolid in MAC—is directly useful to clinicians.
d). Interpretation consistent with current practice: The discussion correctly emphasizes species identification and susceptibility-guided therapy, and notes the practical roles/limitations of macrolides, aminoglycosides (including liposomal amikacin), linezolid, and rifamycins.
Priority revisions (possible)
a). Rewrite the Introduction to be more "brief" and with fewer references.
b). Rewrite the Conclusion to match the journal's guidelines (a concise and objective paragraph addressing the study objective; move any lengthy comments to the Discussion).
c). Harmonization of AST reporting. In each table/figure legend, (i) list n; (ii) specify the CLSI document used; (iii) state whether the amikacin interpretation refers to the intravenous or inhaled liposomal breakpoint, whenever relevant (you already clarified this in the text—reflect in the legends).
d). Limitations paragraph. You note a focused AST subset and species heterogeneity; explain this in the Discussion/Limitations and briefly acknowledge that clinical adjudication of NTM-PD (as per ATS/ERS/ESCMID/IDSA) was out of scope; therefore, these are laboratory surveillance data and not case-level treatment outcomes. (The guidelines are already cited in the reference list.)
e). Minor text edits. Ensure consistent expansion of abbreviations upon first mention in the Abstract and figure legends (e.g., RGM/SGM, MAC).
Further observations and comments were made by this reviewer directly on the text of the attached manuscript.
Best regards,

Author Response
Please see the attachment

Reviewer 2 Report (Previous Reviewer 2)
Comments and Suggestions for Authors
Although nontuberculous mycobacterial (NTM) infections are considered opportunistic, the incidence and prevalence of these infections are increasing globally, posing a significant public health concern. Here the authors provide an up-to-date overview of the epidemiological distribution of NTM in the province of Pavia during the period 2011-2023. The manuscript is well-written, relevant to the subject matter, and presented in a logical and organized manner. The majority of the cited references are recent and relevant. The conclusions are in line with the evidence and arguments provided.
Minor concern.
In the "Study Setting" section, the authors indicate that 456 MTC strains were also incorporated into the analysis. Figures 1 and 2 provide data for these strains. Unfortunately, Table 2 does not provide information on the gender and age distribution of MTC cases compared to NTM cases. It would be useful to know if there are any similarities in the patterns of these cases.
Author Response
Please see the attachment

This manuscript is a resubmission of an earlier submission. The following is a list of the peer review reports and author responses from that submission.
Round 1
Reviewer 1 Report
Comments and Suggestions for Authors
My considerations are in the attached article.
Revise the manuscript with a native English-speaking scientific editor.
Correct all taxonomic and formatting inconsistencies according to Microorganisms guidelines. Rewrite the discussion to include: comparison with at least five recent studies; hypothetical explanations of findings; implications for One Health, epidemiology, or public health. Expand methodology and result interpretation: clarify sampling, diagnostics, and data analysis methods. Ensure all scientific names are current, italicized, and consistent.

Revise the manuscript with a native English-speaking scientific editor.
Correct all taxonomic and formatting inconsistencies according to Microorganisms guidelines. Rewrite the discussion to include: comparison with at least five recent studies; hypothetical explanations of findings; implications for One Health, epidemiology, or public health. Expand methodology and result interpretation: clarify sampling, diagnostics, and data analysis methods. Ensure all scientific names are current, italicized, and consistent.
Reviewer 2 Report
Comments and Suggestions for Authors
Nontuberculous mycobacteria (NTM) are emerging pathogens that are becoming increasingly important in human health. They were initially seen as the main cause of opportunistic infections in people with HIV. But they have also been identified as the root cause of various infections in immune-competent individuals and healthcare-associated infections. NTM can live in the environment and stick to different surfaces. This is important for their ability to cause disease. The point of the study we're revising is to show how the epidemiological landscape of NTM has changed over the last 13 years, from 2011 to 2023, in a specific area of northern Italy.
The research methods employed are appropriate and adequately detailed. A detailed examination of the results is presented in discussion section. The patterns identified in the study are hypothetically explained by the authors. The authors possess a comprehensive understanding of the limitations imposed by the scope of their work.
Minor concern. The investigation of antibiotic resistance patterns could be enhanced through the incorporation of a treatment outcome analysis.